# Computational Investigations of a pH-Induced Structural Transition in a CTAB Solution with Toluic Acid

**DOI:** 10.3390/molecules26226978

**Published:** 2021-11-19

**Authors:** Tingyi Wang, Hui Yan, Li Lv, Yingbiao Xu, Lingyu Zhang, Han Jia

**Affiliations:** 1Technology Inspection Center, Shengli Oilfield Company, SINOPEC, Dongying 257000, China; wangtingyi180.slyt@sinopec.com (T.W.); xuyingbiao.slyt@sinopec.com (Y.X.); zhangly639.slyt@sinopec.com (L.Z.); 2School of Pharmaceutical Sciences, Liaocheng University, Liaocheng 252059, China; 3Changqing Well Technology Work Company, Chuanqing Drilling Engineering Company Limlted, CNPC, Xi’an 710021, China; cqjxlvli123@163.com; 4Shandong Key Laboratory of Oilfield Chemistry, School of Petroleum Engineering, China University of Petroleum (East China), Qingdao 266580, China; jiahan@upc.edu.cn

**Keywords:** molecular dynamics simulation, pH-induced structural transitions, rodlike micelle, sphecial micelle, cationic surfactant

## Abstract

In this work, molecular dynamics simulations were performed to study the pH-induced structural transitions for a CTAB/*p*-toluic acid solution. Spherical and cylindrical micelles were obtained for aqueous surfactants at pH 2 and 7, respectively, which agrees well with the experimental observations. The structural properties of two different micelles were analyzed through the density distributions of components and the molecular orientations of CTA^+^ and toluic acid inside the micelles. It was found that the bonding interactions between CTA^+^ and toluic in spherical and cylindrical micelles are very different. Almost all the ionized toluic acid (PTA^−^) in the solution at pH 7 was solubilized into the micelles, and it was located in the CTA^+^ headgroups region. Additionally, the bonding between surfactant CTA^+^ and PTA^−^ was very tight due to the electrostatic interactions. The PTA^−^ that penetrated into the micelles effectively screened the electrostatic repulsion among the cationic headgroups, which is considered to be crucial for maintaining the cylindrical micellar shape. As the pH decreased, the carboxyl groups were protonated. The hydration ability of neutral carboxyl groups weakened, resulting in deeper penetration into the micelles. Meanwhile, their bonding interactions with surfactant headgroups also weakened. Accompanied by the strengthen of electrostatic repulsion among the positive headgroups, the cylindrical micelle was broken into spherical micelles. Our work provided an atomic-level insights into the mechanism of pH-induced structural transitions of a CTAB/*p*-toluic solution, which is expected to be useful for further understanding the aggregate behavior of mixed cationic surfactants and aromatic acids.

## 1. Introduction

The controllable self-assemblies of the amphiphilic molecules in aqueous solution are hot issues in both scientific and technological areas [1,2]. The size and shape of the surfactant assembly mainly depend on the chemical structures of the surfactants, such as the lengths of the hydrocarbon chains, properties of the polar headgroups, and the counter ions [3,4,5,6]. Generally, surfactants in solution form spherical micelles spontaneously above the critical micelle concentration (CMC) [7,8,9]. With a further increase in surfactant concentration, the spherical micelles may grow into rod-like or wormlike micelles, and even vesicles. 

Adding certain amounts of simple inorganic ions (such as Cl^−^ and Br^−^) or aromatic anions (such as salicylate and benzoic acid) into cationic surfactant solutions can also lead the formation of long rod-like or wormlike micelles at a lower surfactant concentration [9,10,11,12,13]. Due to the stimuli of functional groups in the additive salts, these aggregations consisting of cationic surfactant and anionic additives are sensitive to the external conditions [14,15,16,17,18]. Under an external stimulus, such as pH, temperature, or UV/vis, structural micellar transitions may occur. Thus, controllable self-assemblies of the surfactant in solution can be realized as desired. These stimuli-responsive surfactant systems have attracted much attention in fundamental research and industrial applications, such as drug release, soil remediation, and oilfield industries [1,2].

According to the previous studies on such controllable surfactant systems, it is believed that the interactions between surfactants and the additives are responsive for the stimulus responsiveness of the aggregations. Under external stimuli, no matter what happens to the structure to the surfactant or additive—such as protonated/ionized and cis-transitions—the intermolecular interactions were finally changed. When limited to experimental techniques, it is hard to observe these microscopic interactions directly. Thus, to further investigate the controllable surfactant systems at the molecular level would be significant and useful for understanding the molecular mechanisms behind the specific effects of surfactants or additives on the stimulus-responsiveness performance. 

During the past decades, molecular dynamics (MD) simulations have been proven to be a powerful technique to provide supplemental and microscopic insights into experimental observations [19,20,21,22,23,24]. Many computational studies have been devoted to gain insights into the micro-behavior of the various surfactant systems. However, most previous studies mainly focused on the morphologies of the aggregations. Investigations on the changes in intermolecular interactions inside the aggregations are relatively scarce, especially on the changes in bonding structure induced by external stimuli.

In this work, we studied the structural transitions of a typical cationic surfactant/additive micelle solution induced by pH variation. Cetyltrimethylammonium bromide (CTAB) is one of the most extensively applied cationic surfactants. It forms spherical micelles with a diameter of 2–3 nm when above the CMC in water. These micelles will grow into rod-like or wormlike micelles when the surfactant concentration is far above the CMC (about several hundred times above the CMC). By increasing the ionic strength or adding hydrotrotes into micelle solution, the spherical micelles will undergo a sphere to rod-like shape transition, even at lower concentrations. Besides promoting micellar growth, the aromatic hydrotrotes are sensitive to external conditions, including temperature, UV/vis light, and pH. The aqueous behavior of CTAB in the presence of phenols, salicylate, and aromatic acids has been widely studied [25,26,27,28,29].

The structural transitions of a CTAB/*p*-toluci acid (PTA) micellar solution were investigated as a representative system in this paper. By altering the pH of a CTAB/*p*-toluci acid solution, the surfactants can form micelles with different geometries [3]. Our aim was to study the effects of different intermolecular interactions on the structural transitions of CTAB/*p*-toluci acid aggregations. The simulations started with pre-assembled cylindrical micelles. Experimental observations were successfully reproduced [3]. Based on the MD results, microscopic information on the mechanism behind the pH-induced micellar shape transition has been provided. 

## 2. Results and Discussion

### 2.1. Different Aggregation Morphologies

Figure 1 shows the simulated configurations of the two systems at the beginning and end of the simulations. As expected, spherical micelles were obtained in the presence of the protonated PTA (pH = 2), whereas a rod-like micelle was obtained when all PTA molecules were deprotonated to PTA^−^ (pH = 7). From the final configuration, it can be seen that almost all the PTA^−^ ions were solubilized into the rod-like micelle. In the protonated PTA system, most of the neutral PTA molecules still remained in the water phase. The only two PTA^−^ ions were solubilized into the micelle. Therefore, it is believed the aggregation shape of the micelle should be related to the quantity of the solubilized additives. 

The absolute number of the solubilized PTA^−^ into the rod-like micelle was counted as a function of simulation time, as shown in Figure 2. In addition, the radius of the rod-like micelle with time evolution was monitored to show the changes that solubilization brought to the micellar shape. The radius of the rod-like micelle was defined by the average distance between N atoms and the central axis of rod-like micelle. In the initial configuration, the surfactants were loosely packed, yielding a large radius (~2.4 nm) of the pre-assembled micelle. As the simulation went on, the rod-like micelle showed great fluctuation. Meanwhile, the pre-assembled micelle began to shrink due to the hydrophobic interactions between the surfactant chains. A great deal of the PTA^−^ ions began to enter into the CTA^+^ aggregation. At about 7.5 ns, the solubilized numbers of PTA^−^ gradually reached stable values. Subsequently, the fluctuation on the micelle gradually disappeared, resulting in a stable and rigid long rod-like micelle. The radius of the micelle also reached a constant value of about 1.95 nm. The stable aggregated structure indicated the simulation system reached equilibrium, so a total simulation time of 20 ns was sufficient.

### 2.2. Detailed Structural Properties of the Formed Micelles

As discussed above, the structural transition with the variation in pH is related to the solubilization of the additives into the micelle. Thus, the interactions between additives and surfactants play an important role in stabilizing the micellar structure. Before discussing the intermolecular interactions between additives and surfactants, we must first investigate the distribution of these hydrotropes inside the micelle. The locations of some selected species were characterized by calculating the number density distribution profiles. In Figure 3, the number density distributions were plotted with respect to the central axis of the rod-like micelle, which is along the z-axis of the simulation box. For the spherical micelle, the number density was calculated with respect to the center of mass (COM) of the spherical micelle, i.e., along the radial direction of the spherical micelle. In the simulated system with protonated PTA (pH = 2), three spherical micelles were obtained at the end of the simulation, as shown in Figure 1. We selected the biggest one to calculate the structural properties.

The results for both the rod-like and spherical micelles are very similar to those of the previous simulation studies [30,31]. It can be seen that the surfactant headgroups which were presented by headgroup N atoms constituted a shell region the surface of the micelle, and the hydrophobic chains were concentrated in the interior in both rod-like and spherical micelles. Our focus is the distribution of the additive molecules PTA or PTA^−^ inside the micelles. It was found that the terminal methyl groups of PTA or PTA^−^ were located deeply in the hydrophobic region in both rod-like and special micelles. The carboxyl groups were located on the outer shells of the micelles, and they were adjacent to the surfactant headgroups. This was certainly because that the phenyl groups were hydrophobic and the carboxyl groups were hydrophilic. There are mainly two differences between PTA and PTA^−^. One is that the distributions of the two carboxyl O atoms in deprotonated PTA^−^ ions overlapped, suggesting they were distributed at the same locations inside the rod-like micelle. The peaks in the distributions of the two carboxyl O atoms in protonated PTA^−^ molecules are staggered. The protonated O2 atoms were located outside a little bit more than the other O1 atoms. The other difference is that the distance between COO^−^ O atoms and headgroup N atoms was quite short (~0.1 nm), which was measured by the two distribution peaks shown in Figure 3, whereas the distances between COOH O atoms and headgroups N atoms were rather long (~0.7 nm).

The above results show that once the carboxyl groups were protonated with a decrease in pH, the PTA molecules localized more deeply inside the interior of the spherical micelle. This suggests that when the carboxyl groups are changed to be electroneutral, the hydrophobic interactions between methylbenzene groups and CTA^+^ hydrocarbon chains will ultimately dominate. The O2 atom in COOH group being located outside a little bit more was mainly due to the stronger interactions between hydroxy groups and water molecules, whereas in the rod-like micelle, the surfactant headgroups were close to the COO^−^ groups, suggesting strong intermolecular interactions through electrostatic interactions. It is believed that the tight bounding between ammonium groups and COO^−^ groups plays an important role in maintaining the cylinder micellar shape.

### 2.3. Bonding Structures of PTA^−^/PTA and Surfactants

The detailed interactions between the additive molecules with the surfactants were further investigated by exploring the orientations of PTA^−^/PTA inside the micelle. The orientation was defined by the angle *θ* between the molecular axis of PTA^−^/PTA and CTA^+^. The molecular axes of PTA^−^/PTA and CTA^+^ were defined by the vector C8 to C1 (atoms in PTA) and the vector C3 to N (atoms in CTA^+^). When calculating the angle, only the neighboring pairs of PTA^−^/PTA and CTA^+^ molecules were considered; i.e., only the interactive pairs which were judged by their separation distances were counted.

The probability distributions of the angles for PTA^−^ and PTA are shown in Figure 4. It is evident that the molecular axis of ionized PTA^−^ preferred to form an angle of about 20° with its adjacent surfactant molecules. When the ionized PTA^−^ ions were protonated, the distribution of angle between the same vectors became very broad. It can be seen that the value of angle varied from 20° to 90°, suggesting the protonated PTA molecules did not prefer to form certain angles with the surfactants. Figure 4 shows the selected bonding structures between PTA^−^/PTA and CTA^+^. Obviously, the ionic PTA^−^ interacted with neighboring CTA^+^ through electrostatic interactions between their carboxyl and ammonium groups. The strong electrostatic interactions resulted in tight bonding between PTA^−^ and CTA^+^. While the PTA^−^ ions were protonated, the strong electrostatic interactions with CTA^+^ surfactants disappeared. Therefore, the bonding between surfactants and additives also weakened inside the aggregates, which is considered to be essential for the shaper transition of the micelle.

### 2.4. Intermolecular Interactions

As discussed above, the bonding mode between surfactants and additives may have an influence on the micellar shape. In addition, the surrounding water solution environment may also affect the interior intermolecular interactions. In what follows, some special intermolecular interactions in two micellar systems were investigated to explore the micro-mechanism behind the micellar shape transition induced by pH variation.

First, the intermolecular interactions between PTA/PTA^−^ and CTA^+^ were visualized by analyzing the weak interactions using the Multiwfn software [32]. The reduced density gradient (RDG) was plotted as a function of electron density ρ(r) based on the selected configurations. The gradient isosurfaces were then visualized with the VMD software [32] to show representations of the weak interactions. As shown in Figure 5a, distributions colored in dark blue present interactions between an ionic CTA^+^ headgroup and PTA^−^, which correspond to the strong attractive interactions. The attractive interactions were mainly attributed to the electrostatic attraction, wheres, the interaction region between CTA^+^ and the neutral PTA disappeared. Instead, weak hydrogen bonds may exist between carboxy group and hydrogen atoms in CTA surfactant.

The hydration effect of surfactants and additive PTA/PTA^−^ was then investigated through the radial distribution functions (RDFs). Figure 6a shows the RDFs of water molecules around the carboxyl groups in PTA or PTA^−^. As shown in RDF profiles, we can see that there were two well-defined hydration shells around the PTA^−^ carboxyl groups, suggesting ordered arrangement of water molecules around carboxyl groups. The high intensity of the first peak demonstrates strong interactions between the ionized carboxyl groups and water molecules. This kind of interaction fell off rapidly when the ionized carboxyl groups were protonated. Therefore, the oxygen atoms in carboxyl groups of natural PTA molecules were located deeper inside the micelle, as shown in Figure 3.

Figure 6b shows the RDFs between surfactant CTA^+^ headgroup N atoms, which can be used to reflect the aggregating degree among the surfactant headgroups. It can be seen that there were two evident aggregated peaks around surfactant headgroups in the rod-like micelle with the ionized PTA^−^. The first peak at about 0.6 nm in its RDF represents the nearest headgroups around one central CTA^+^ headgroup, and the second peak at about 0.8 nm represents the headgroups located at the outer shell. In Figure 7, the aggregated structure of the surfactant headgroups in the rod-like micelles is highlighted to show the detailed information. Due to the tight bonding between CTA^+^ and PTA^−^ through electrostatic interactions, the electrostatic repulsion among the positive headgroups was effectively weakened. The electrostatic shielding among the headgroups introduced by PTA^−^ is therefore considered to play an essential role in maintaining the structure of the rod-like micelle.

For the spherical micelle in the presence of PTA, it is evident that the interactions among the surfactant headgroups weakened greatly. As can be seen from Figure 6b, there was only a shoulder peak at about 0.6 nm. This suggests that the surfactant headgroups were loosely packed, compared with those in the rod-like micelle (Figure 7). Due to the disappearance of the electrostatic shielding from the additive molecules, the positive CTA^+^ headgroups repelled each other. Cooperating with the hydrophobic interactions from the surfactant tails, the aggregations prefer to form spherical micelles.

## 3. Computational Details

First, according to the previous studies [23,24,30,31], a pre-assembled cylindrical micelle was built. The obtained cylindrical micelle consisting of 180 CTA^+^ surfactants was placed in a simulation with dimensions of 25 nm × 25 nm × 10 nm. The central axis of the cylindrical micelle was placed centrally in the box along the z direction of the simulation box. Based on the experimental conditions [3], two systems were simulated to investigate the micellar shape transitions induced by pH. The first system was constructed by inserting 90 PTA^−^ molecules around the pre-assembled micelle, to study the micro-behavior of a CTAB/PTA^−^ solution at pH 7. The second system corresponded to the situation at pH 2. The acidic environment was represented by adding certain amounts of hydronium and chloride ions. As usual, the hydronium ions were in their hydrated ion forms (H_3_O^+^). In the acidic situation, 88 PTA^−^ ions were protonated according to the p*Ka* value of benzoic acid at 298 K. Finally, bromide ions were inserted into the above two systems and the simulation boxes were filled with water molecules. The compositions of two systems are summarized in Table 1. 

Molecular dynamics simulations were performed using the Gromacs package (version 2019.3) [33,34,35,36]. The united-atom GROMOS 54A7 force field [37] was used to describe the intermolecular interactions. Structures of the surfactant and additives are shown in Figure 8. The force filed parameters for the molecules, including CTA^+^, PTA/PTA^−^, and H_3_O^+^, were obtained using the Automated Topology Builder (ATB) server [38]. Water molecules were described by the simple point charge/extend (SPC/E) model [39]. The two systems were first minimized through the steepest descent method. Then, a 20 ns MD simulation under the NPT ensemble was performed for each system. During the simulation, the temperature (298 K) and pressure (1 atm) were maintained by the V-rescale thermostat and Berendsen barostat with coupling time constants of 01. and 1.0 ps, respectively [40,41]. LINCS algorithm [42] was applied to constrain the bond lengths of other components. Periodic boundary conditions were applied in all three directions. The cut-off distance for the Lennard–Jones and electrostatic interactions was 1.2 nm. The particle mesh Ewald method was used to calculate the long-range electrostatic interactions [43]. Configurations were visualized using Visual Molecular Dynamics software [44].

## 4. Conclusions

Molecular dynamics (MD) simulations were performed to investigate the pH-induced structural transitions in aqueous CTAB/PTA solutions. Two simulated systems were created. One was a system consisting of CTAB and neutral PTA, which represented the solution in an acidic environment (pH = 2). The other system consisted of CATB and ionized PTA^−^ (pH = 7). The two systems were both simulated using a pre-assembled cylindrical micelle. The MD results reproduced the experimental phenomenon—that is, spherical and rod-like micelles were obtained for the systems at pH 2 and 7, respectively. The mechanism behind the pH-induced micellar shape transitions was investigated on the basis of the MD results. It was found that the ionized PTA^−^ can effectively screen the electrostatic repulsions among the positive surfactant headgroups, through the strong interactions with surfactant headgroups. The dense packing of the surfactant headgroups lead the formation of a rod-like micelle. With the lower pH, the ionized carboxyl groups were protonated. The bonding of the neutral PTA with surfactant weakened, resulting in the strengthening of electrostatic repulsion among surfactant headgroups. The loose packing among surfactant headgroups resulted in breaking of the cylindrical micelle and the formation of the spherical micelles. Our study provided a molecular mechanism for the pH-induced shape transition in a mixed cationic surfactant and aromatic ions solution. The results presented intuitionistic intermolecular interactions which were responsible for the micellar shape transition. These observations are expected to be useful for the environmental stimuli-responsive colloid systems in experimental studies.

## Figures and Tables

**Figure 1 molecules-26-06978-f001:**
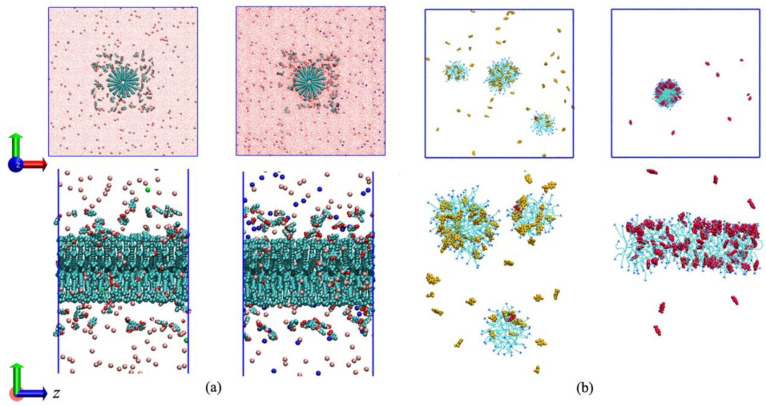
(**a**) Initial setup of the two simulated systems. (**b**) Configurations of CTAB/PTA (pH 2, left in inset a and b) and CTAB/PTA^−^ (pH 7, right in inset a and b) aggregations. The top panel shows the views from the z-axis perspective. The solid blue lines represent the periodical boundary conditions. Water molecules and inorganic ions are hidden in inset b for clarity. The neutral forms PTA and ionized PTA^−^ are displayed in yellow and red, respectively.

**Figure 2 molecules-26-06978-f002:**
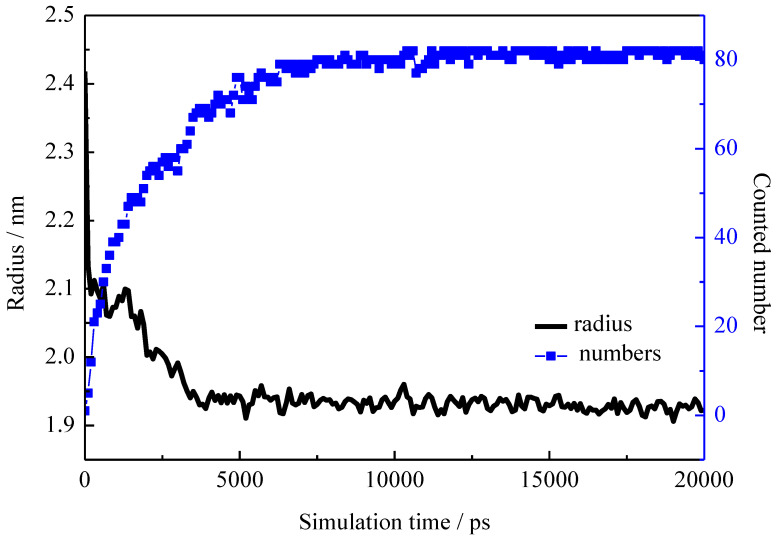
Radius of the rod-like micelle and solubilized numbers of PTA^−^ in the micelle plotted as a function of simulation time.

**Figure 3 molecules-26-06978-f003:**
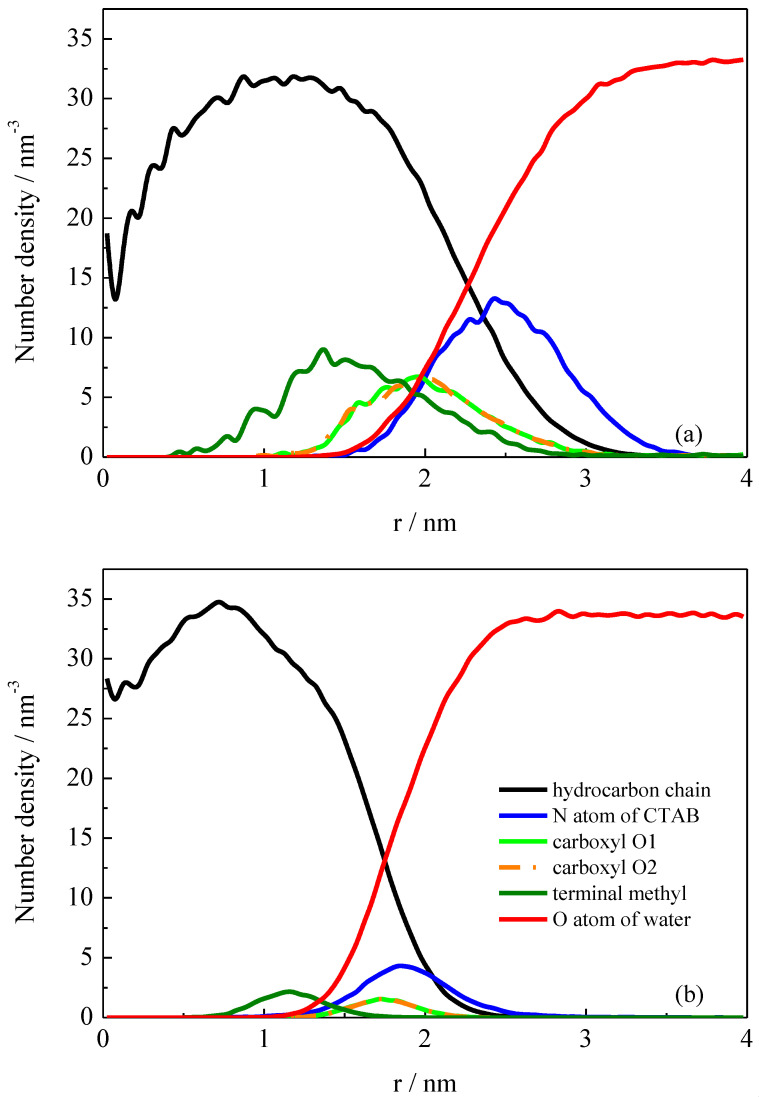
(**a**) Number density distributions of components with respect to the COM of the spherical micelle. Values for N1, O1, O2, and C8 (see Figure 8) were increased 10 times for clarity. (**b**) Number density distributions of components with respect to the central axis of the rod-like micelle.

**Figure 4 molecules-26-06978-f004:**
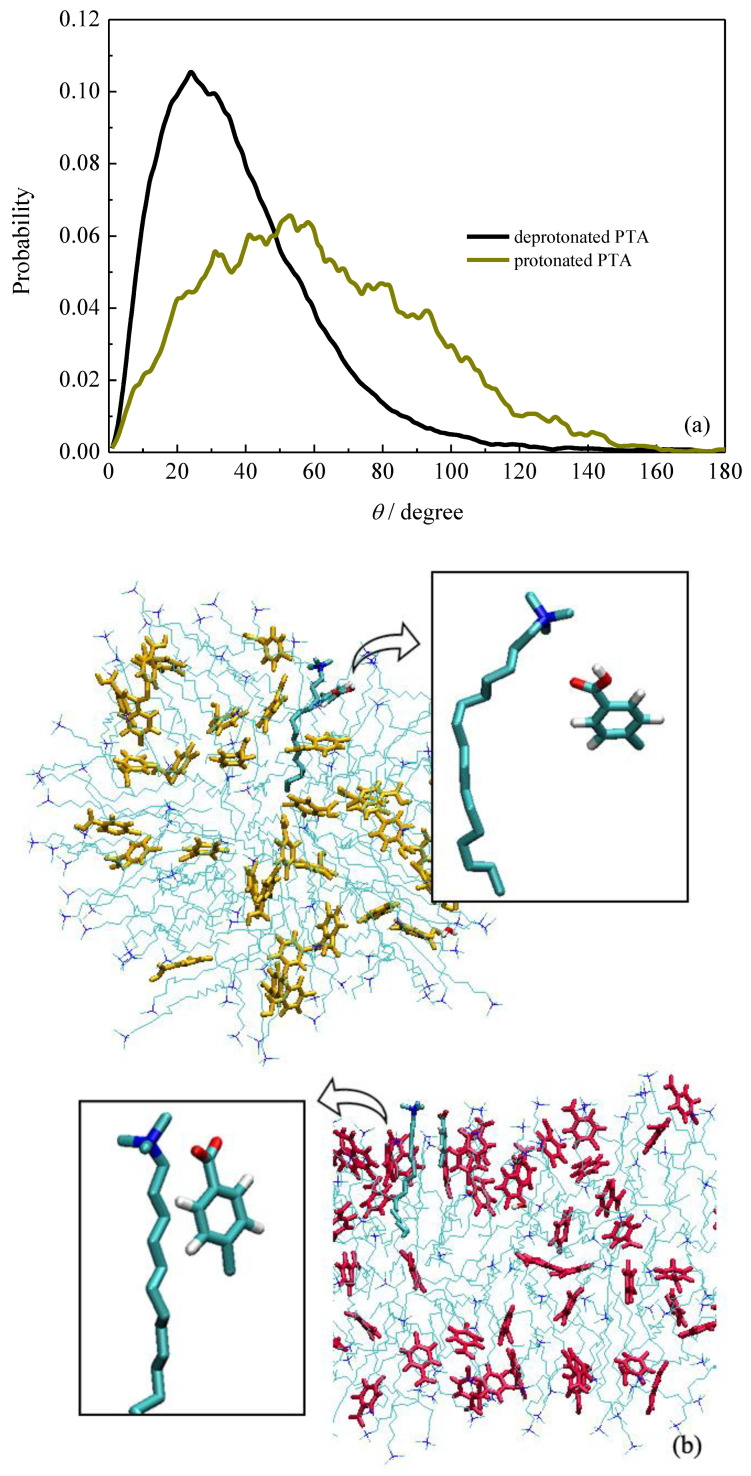
(**a**) Probability distribution of the angle between vectors defined in the molecular structures. (**b**) Bonding structures between CTA^+^ and PTA in spherical and rod-like micelles.

**Figure 5 molecules-26-06978-f005:**
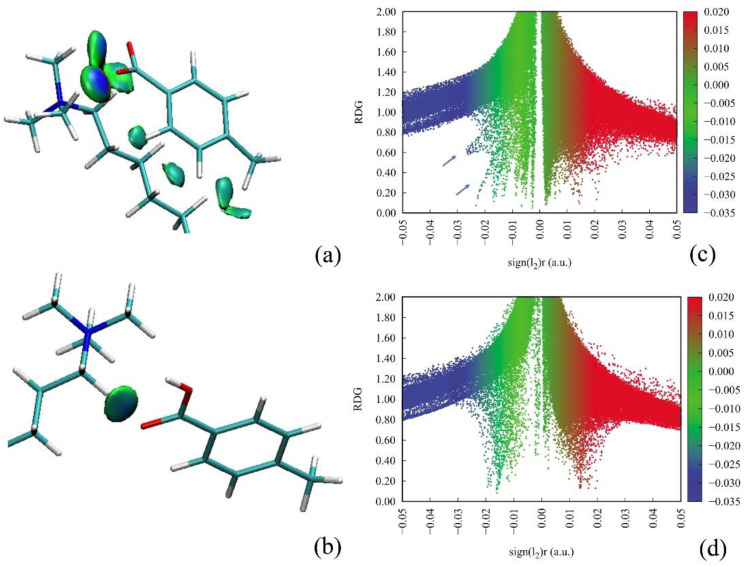
Weak interaction analysis for (**a**) PTA^−^/CTA^+^ and (**b**) PTA/CTA^+^. Insets (**c**,**d**) show the reduced density gradient (RDG) versus electron density for configurations shown in insets a and b, respectively.

**Figure 6 molecules-26-06978-f006:**
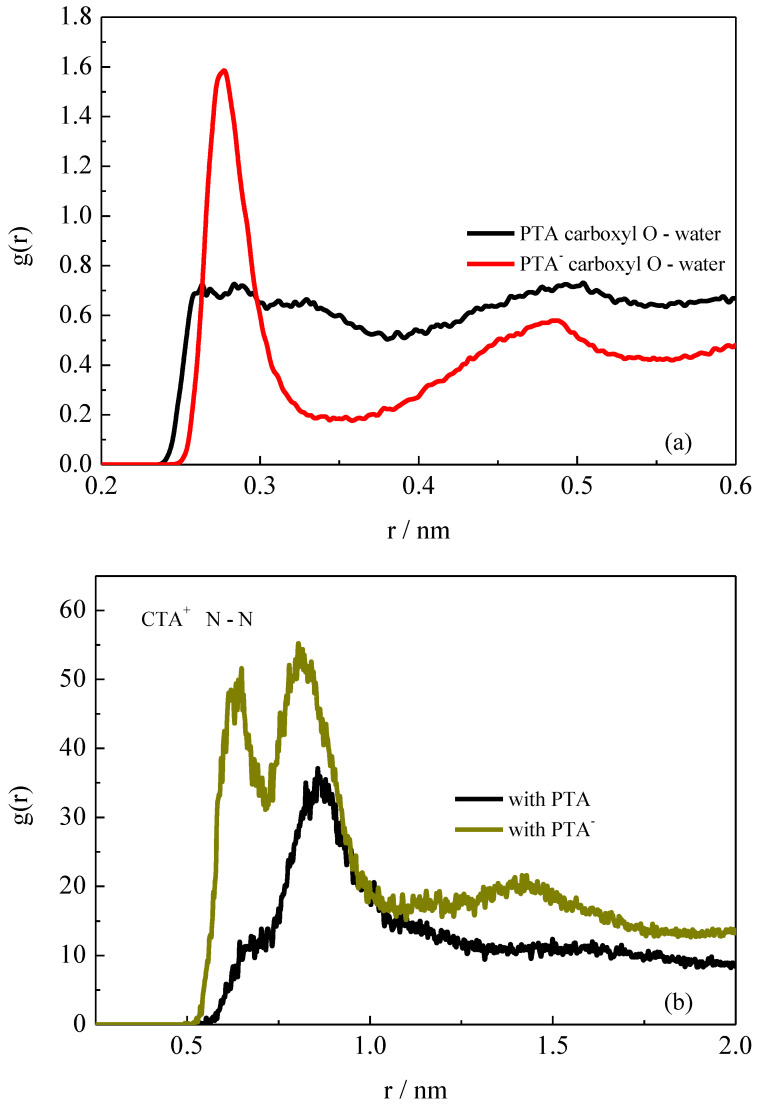
(**a**) RDFs between surfactant N atoms and water O atoms. (**b**) RDFs between surfactant N atoms.

**Figure 7 molecules-26-06978-f007:**
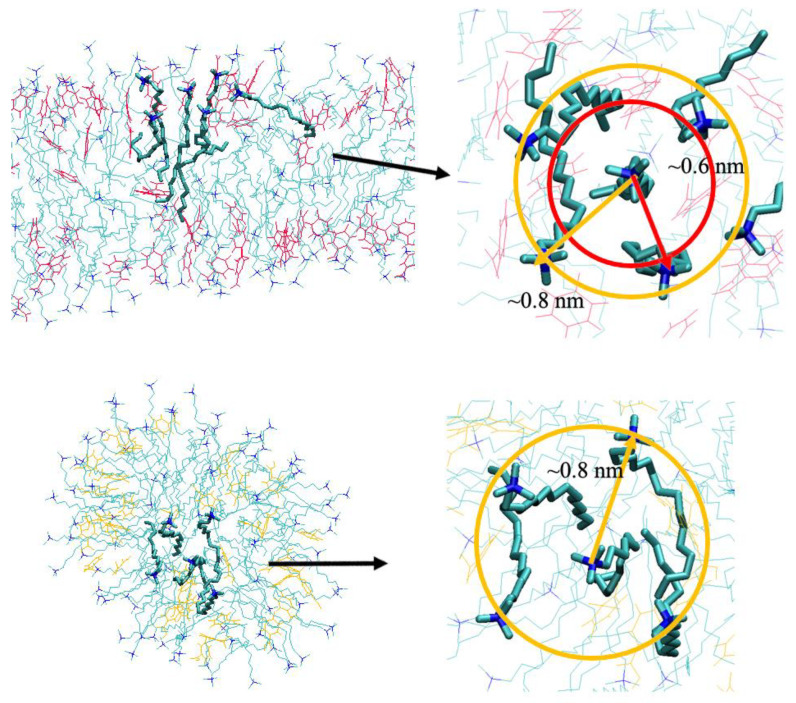
Packed structures of surfactants inside rod-like (**top panel**) and spherical (**bottom panel**) micelles.

**Figure 8 molecules-26-06978-f008:**
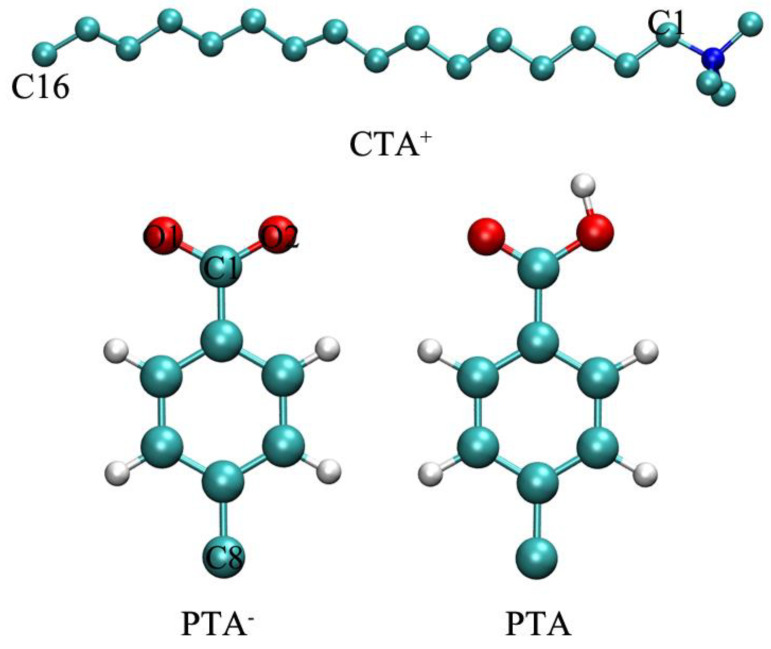
Structures of CTA^+^, PTA^−^, and PTA. (Atomic color scheme: C, cyan; O, red; N, blue; and H, white).

**Table 1 molecules-26-06978-t001:** Simulated systems: numbers of each component in the different systems.

Scheme 2.	CTA^+^		Br^−^	PTA	PTA^−^	H^+^	Cl^−^	Na^+^	Water
pH 2	180		180	88	2	9	7		193623
pH 7	180		180		90			90	193636

## Data Availability

The data presented in this study are available on request from the corresponding author.

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
