# Peer review of "Computational Investigations of a pH-Induced Structural Transition in a CTAB Solution with Toluic Acid"

_molecules, 2021, doi:10.3390/molecules26226978_

Round 1
Reviewer 1 Report
Manuscript entitled “Computational Investigations of a pH-Induced Structural Transition in CTAB Solution with Toluic Acid” written by Wang et al. treats about molecular dynamics calculations of structural reorganization for Cetyltrimethylammonium bromide (CTAB)/p-toluic acid (PTA) solution induced by pH changes. As the examined system is a surfactant, the micellar architectures in solution were analysed. With neutral PTA (acidic solution) spherical micelles were produced while with anionic PTA (pH=7) the rodlike ones were obtained. The mechanism governed these shape changes were thoroughly evaluated by MD methods. This work brings interesting results into the field of surfactant chemistry and is technically correct, well-written with clear, coherent style. Therefore, I recommend to publish this material nearly as it stands with minor comments from my side:
- In Table 1 why the number of CTA+ molecules differs between two approaches? 180 (at pH=2) and 108 (at pH=7)?
- At the end of conclusion section, a short perspective should be given on the potential of using the Authors’ results in practical application.
- Is it possible to visualize even one or couple electrostatic interaction in systems discussed? I mean, is it possible to show exemplary direct bonding path between the atoms engaged?
- Some references require reformatting, like unification of the font (bolding or unbolding year; volumin numbers in italics or not, e.g. ref. 7-9, 27, 28, etc.)
Reviewer 2 Report
The paper is a very interesting application of the MD simulation in the description of the intermolecular forces between surfactant molecules, that form spherical and/or cylindrical micells in accord to the pH.
The MD simulation were done properly and the obtained data sound solids, also their analysis were properly done.
The conclusion is appropiate.
I suggest the author the following few modifications:
- Please, update figure 2 showing both the initial (t =0) and final configuration (t= 20 ns) of these systems at pH 2 and 7.
- Please, check carefully the use of abbreviations in both the abstract and the text. Sometime the ionized p-toluic acid is wrongly indicated PTA+ (lines: 19, 178, 215, 243)
- Please, check carefully the English of the manuscript, several mistakes are evident ( lines: 58, 61, 74, 76, 78, 85,144)
- Please, check for not properly written sentences (137-138, 140, 173-174, 179, 252)
- Line 239, please, replace 'Figure 6c' with 'Figure 7'
Author Response
Please kindly check the file attached.
